# Characterization of structural changes in modern and archaeological burnt bone: Implications for differential preservation bias

**Giulia Gallo**[1,2]*, **Matthew Fyhrie**[1], **Cleantha Paine**[3], **Sergey V. Ushakov**[4], **Masami Izuho**[5], **Byambaa Gunchinsuren**[6], **Nicolas Zwyns**[1,2,7], **Alexandra Navrotsky**[4]

1 Department of Anthropology, University of California Davis, Davis, CA, United States of America, 2 Center for Experimental Archaeology at Davis, University of California Davis, Davis, CA, United States of America, 3 Archaeology Institute University of the Highlands and Islands, Kirkwall, United Kingdom, 4 School of Molecular Sciences and Center for Materials of the Universe, Arizona State University, Tempe, AZ, United States of America, 5 Tokyo Metropolitan University, Tokyo, Japan, 6 Institute for History and Archaeology, Mongolia Academy of Science, Ulaanbaatar, Mongolia, 7 Department of Human Evolution, Max Planck Institute for Evolutionary Anthropology, Leipzig, Germany

* gtgallo@ucdavis.edu

**Data Availability Statement:** All relevant data are within the manuscript and its Supporting Information files.

## Abstract

Structural and thermodynamic factors which may influence burnt bone survivorship in archaeological contexts have not been fully described. A highly controlled experimental reference collection of fresh, modern bone burned in temperature increments 100–1200°C is presented here to document the changes to bone tissue relevant to preservation using Fourier transform infrared spectroscopy and X-ray diffraction. Specific parameters investigated here include the rate of organic loss, amount of bone mineral recrystallization, and average growth in bone mineral crystallite size. An archaeological faunal assemblage ca. 30,000 years ago from Tolbor-17 (Mongolia) is additionally considered to confirm visibility of changes seen in the modern reference sample and to relate structural changes to commonly used zooarchaeological scales of burning intensity. The timing of our results indicates that the loss of organic components in both modern and archaeological bone burnt to temperatures up to 700°C are not accompanied by growth changes in the average crystallite size of bone mineral bioapatite, leaving the small and reactive bioapatite crystals of charred and carbonized bone exposed to diagenetic agents in depositional contexts. For bones burnt to temperatures of 700°C and above, two major increases in average crystallite size are noted which effectively decrease the available surface area of bone mineral crystals, decreasing reactivity and offering greater thermodynamic stability despite the mechanical fragility of calcined bone. We discuss the archaeological implications of these observations within the context of Tolbor-17 and the challenges of identifying anthropogenic fire.

## Introduction

Anthropogenic burnt bones can be indicative of many social and economic behaviors and can contribute to studies identifying evidence for ritual activity [1], cremations [2–4], bone fuel

**Funding:** NZ, Grant #156074, National Science Foundation (https://www.nsf.gov/awardsearch/showAward?AWD_ID=1560784). MI PaleoAsia Project Grant No. 1802, FY2016–2020) from the Ministry of Education, Culture, Sports, Science and Technology, Japan ("Cultural history of PaleoAsia: Integrative research on the formative processes of modern human cultures in Asia," directed by Yoshihiro Nishiaki) (http://paleoasia.jp/en/). GG UC Davis Cluster Grant "Archaeology and Soil Science Synergy" (https://dhi.ucdavis.edu/events/2019-2020-dhi-transcollege-research-clusters-call-proposals). The funders had no role in study design, data collection and analysis, decision to publish, or preparation of the manuscript.

**Competing interests:** The authors have declared that no competing interests exist.

[5–9], hygienic practices [5, 9], cooking and marrow warming [10], and locations of combustion features [11]. Burnt bones undergo substantive structural and compositional changes at different burning intensities, however, and the implications of these changes for differential bone survivorship in an archaeological fauna assemblage is of critical importance for studies utilizing burnt material.

Bone diagenesis can result in bone mineral disintegration or dissolution [12]. The rate, sequence, and extent of diagenetic processes are determined by many factors, including the nature of the depositional environment and the age, element, and species of the bone tissue. Postmortem bone preservation is well described in regards to the differences between compact and cancellous bone [13–16], juvenile and adult bone [17, 18], intra- and interspecies variation in bone size and density [19–21], and different environmental conditions [12, 22–26]. However, zooarchaeological evaluations of burnt bone preservation have resulted in varied interpretations and have not addressed the different structural properties of bone burnt to different temperatures [5, 27].

The aim of this study is to describe the range of structural modifications to bone mineral produced by burning at different temperature intensities, and to relate such changes to standardized scales utilized in zooarchaeological methods, specifically here the Stiner et al. [27] scale of burning intensity. This is done with the intention of describing differences between categories of burnt bone, including any vulnerabilities to diagenetic processes that could result in assemblage biases within burnt bone in archaeological contexts. Here we present the results of a controlled experimental reference library of fresh, modern bone burnt in increments of 100–1200°C and analyzed with Fourier transform infrared spectroscopy (FTIR) with Attenuated Total Reflectance (ATR) attachment and X-ray diffraction (XRD). We additionally compare spectroscopic measurements to a sample of burnt fauna dated to ca. 30 ka to verify visibility of alterations in an archaeological assemblage.

## Bone

Bone is comprised of organic proteins primarily of collagen, inorganic mineral, and water, creating a composite material organized in compact bone in cylindrical structures of concentric lamellae surrounding an interior channel for a central blood canal [28, 29]. This hierarchical arrangement provides and maintains the biological roles of skeletal tissue: mechanical strength to transmit force and protect organs, and the regulation of homeostasis through ionic regulation [28–31].

Living bone is very porous, with around 12% of bone volume comprised of open spaces [2]. The concentric systems, known as Haversian systems, constitute a large percentage of the bone matrix porosity, with the remainder composed of resorption bays and voids created between the organic and inorganic components [29–32]. The amount, size, and density of pores in bone is variable across elements, species, and ages, although trabecular bone does exhibit a higher porosity than compact bone due to its more open structure [21, 29, 32].

Inorganic bone constituent, bioapatite, is isostructural to mineral hydroxyapatite $Ca_5(PO_4)_3OH$. The specific chemical compositions of bioapatite reflect diet, biological age through history of bone remodeling, and variation can exist both within species and within the skeletal elements themselves [29]. Bioapatite contains 5–8 wt% carbonate which can substitute either phosphate or hydroxyl group in hydroxyapatite structure [33]. Bioapatite has a high degree of nonstoichiometry, and its composition can be described as $Ca_{10-x}(PO_4)_{6-x}(HPO_4, CO_3)x(OH,1/2CO_3)_{2-x}$ with $0<x<2$ [31, 34, 35].

*In vivo* bioapatite has extremely small, thin, plate-like morphologies (1–7 nm thick, 15–200 nm in length, and 10–80 nm in width) which are cross-linked to organic collagen fibrils [34–

36]. Water is found in bone as loose mobile water in the extracellular matrix, in void spaces to facilitate movement, and integrated within and around the organic and mineral components [31, 37]. Bioapatite crystallites have typical surface areas above 200 m$^2$g and are heavily hydrated [30, 31, 35–38]. These surface layers of ions play a key role in the regulation of homeostasis, as they can be easily exchanged and provide a necessary capacity to regulate ionic concentrations in living tissue [3, 28, 38–41].

## Bone diagenesis

Diagenesis, the postmortem changes to bone tissue in burial environments, includes the integrated processes of microbial attack, water activity, and mineral recrystallization and can result in the complete disintegration of bone material [12]. The arrangement and size distribution of pores at the time of burial are large predictors of bone decay or bone survival, as pores mediate the access and extent of destructive agents such as bacteria and water [18, 22–24, 42–44]. Microbial attack itself is an active and immediate process accounting for a large amount of initial organic destruction, especially in warmer environments [42]. Microbial access to collagen degrades the protein chains, effectively removing the organic component of bone [19, 24, 32, 42, 44, 45]. The removal of the collagen component results in a more brittle biomaterial on the macroscale, and leaves bioapatite crystals unprotected on the micro- and nanoscale [46].

Exposed bioapatite is vulnerable to the incorporation of impurities and to disintegration, as postmortem crystals initially retain the specific morphology, reactivity, and thermodynamic instability of living bioapatite [47, 48]. The reaction between bone mineral and water is the most significant predicate of bioapatite disintegration at this stage of diagenesis, and bones buried in environments with active water movement are highly vulnerable to leeching and dissolution, noted to be heightened when bioapatite is exposed and easily accessible after organic removal [24, 48]. There is no universal thermodynamic model of bioapatite solubility due to the complexity of bone as a biomaterial, and rather each crystal domain is assigned its own Metastable Equilibrium Solubility (MES): a distribution phenomenon dependent on aspects of bone quality such as carbonate substitutions, ion vacancies, low crystallinity, and small crystal sizes [48, 49]. Uptake of contamination from the burial environment, such as rare earth elements and secondary calcite, has been noted to reduce as bone mineral spontaneously recrystallizes without in-vivo regulation and larger crystals grow at the expense of smaller crystals [18, 50, 51]. This process results in bone mineral with a slightly higher crystallinity, effectively decreasing the available reactive surface area of bioapatite and therefore the overall solubility compared to fresh bone [18, 36, 50, 51].

## Burnt bone

Burning bone results in the decomposition of the organics and loss of water, as well as in massive changes to bioapatite crystal dimensions and structure. The extent and degree of these alterations are correlated to temperature and burning atmosphere, producing bones with different mechanical and thermodynamic properties dependent on the extent of burning. These micro- and nano- scale transformations have a notable impact on visible macroscopic changes to heat altered bone, including color changes, cracking, shrinkage, weight loss, and fragmentation [27, 41, 52–55]. Burnt bone coloration is generally correlated to burning intensity, and the ease of color identification has assisted in the proliferation and use of zooarchaeological scales of coding heat alteration, such as the Stiner et al. [27] classification of burning intensity (Table 1).

Scales of burning based on macroscopic visual cues are tremendously beneficial for processing archaeological assemblages of burnt fauna, but do not reflect the sequence of changes in

**Table 1. Burning intensity scale based on macroscopic visual qualities following Stiner et al. [27].**

| Burning Scale | Description |
|:---:|:---|
| 0 | Not burnt |
| 1 | Slightly burnt, $< 50\%$ carbonized |
| 2 | Majority burnt, $> 50\%$ carbonized |
| 3 | Fully carbonized |
| 4 | Slightly highly burnt, $< 50\%$ calcined |
| 5 | Majority highly burnt, $> 50\%$ calcined |
| 6 | Fully calcined |

the composition and structural properties of burnt bone. Observations on the nano- and micro-scale have therefore led to the definition of four stages of burning which are correlated to the transformation of bone mineral and removal of organics on the nano- and microscale: dehydration, decomposition, inversion, and fusion [2, 54–57]. These stages are accomplished at different temperature thresholds and were defined in oxidizing burning conditions [2, 54–57]. The rate and degree of temperature induced changes depend on variables such as flesh coverage, heating and cooling rates and oxygen availability [54, 55]. Dehydration, or, the loss of water, occurs between 100 and 600˚C [2, 54, 56–58]. This wide temperature range likely accounts for the quicker loss of the loosely bound water between 25 and 250˚C and the eventual loss of the additional water more structurally bound to the mineral in temperatures above 100˚C [2, 31, 56, 57].

After initial dehydration, the second stage of bone combustion is organic decomposition, from 300 to 800˚C [2, 56, 57]. With collagen degradation starting at 112–260˚C, above 300˚C a large proportion of the organics is reduced to a char [58]. Between 300 and 500˚C most mass, 50–55%, is lost, and above 500˚C any remaining char is removed by 700˚C [58]. The macroscopic transformation most noticeable with the decomposition stage is the striking changes in coloring, with bone becoming visibly blackened with the charring of organics (300˚C), corresponding to Stages 1–3 of the Stiner et al. [27] scale, and after the complete removal of organics (700˚C) transitioning to a grey and chalky white hue for Stiner et al. [27] Stages 4–6 [2, 27, 54, 56, 57]. Bone that is blackened is referred to as combusted or carbonized dependent on burning atmosphere, while grey and white bone with all organics removed can be referenced as calcined [27, 54–56].

Simultaneous to the loss of organics is the alteration of the bioapatite mineral, or the inversion stage, between 500 and 1100˚C [2, 57, 58]. With the removal of the organic component at 300˚C, the larger, plate-like crystals can spontaneously grow at the expense of smaller crystals [2, 57–59]. Experiments with bone burnt while powdered and subsequently cleaned with acetone report mean crystallite size increasing to 10–30 nm, and crystallite thickness moving from 2 to 9 nm [59, 60]. Above 500˚C, additional growth has been observed, with reported crystallite sizes plateauing at 110 nm and with crystal thickness reaching 10 nm [59]. The crystals, transforming from platelet like to hexagonal, later become equiaxed at 900˚C, growing more spheroidal with overall dimensions reaching 300–550 nm [60].

The last stage of heat alteration to bioapatite, fusion, accounts for the microstructural changes noted with the inversion phase above 700˚C [2, 57, 58]. Bone porosity initially increases from the originally porous *in vivo* status with the loss and charring of organics (~300˚C), which also corresponds to a loss in bone density [32, 60]. Carbonized and charred bone is reported to be most porous right before temperatures of calcination (600˚C) [32]. Beginning at 700˚C there is a densification as the bioapatite crystal grains grow, and by 900˚C

there is a total structural coalescence from the additional crystal growth, resulting in an interlocking structure and a marked decrease in porosity [2, 59, 60].

These changes are all products of burning in oxidizing conditions [55, 56]. If a bone is brought to temperatures greater than 300˚C without access to oxygen, a different pattern of thermal alteration has been demonstrated in controlled experiments [55]. When heating occurs in reducing atmospheres, the organic char is not removed and instead becomes more ordered [55]. The crystallinity of the bioapatite does increase, however, although at a slower rate than indicated in oxidizing conditions [55]. New compounds, such as cyanamide, are also likely formed around 600˚-700˚C [55]. Bones burnt in reducing atmospheres above 600˚C do not lose the organic char component, and therefore remain black in coloring [55].

### Burnt bone diagenesis

The rapid morphological and compositional changes to burnt bone tissue are similar to changes seen over prolonged periods of time in the diagenesis of unburnt bone. This includes the removal of organic components and incorporated water, as well as the recrystallization of the bioapatite crystals. The immediate and greater extent of these changes in bone burnt to both low and high temperatures, however, results in a markedly different biomaterial at time of burial than unburnt bone.

Burnt bone is more fragile than unburnt bone, with fragmentation a function of burning intensity [27]. The dehydration and eventual complete removal of collagen from bone tissue significantly changes the toughness and strength properties of bone, altering the density, the structural integrity, and the stress and strain relationship [27, 61–63]. This ultimately results in a greater likelihood of mechanical fracture correlated to the amount of collagen lost, leaving calcined bone the most mechanically vulnerable [27, 61]. Due to this extreme friability, recovered burnt bone fragments do not reflect initial size at deposition and processes such as burial and trampling can severely and easily fragment burnt bone [27, 64].

The fragility, likely presence of small fragment sizes, and elimination of organic components of bone burnt to lower temperatures provides greater surface area and easy access for diagenetic agents in the context of burial environments. Bone mineral does, however, undergo tremendous crystallite growth and reorganization with burning at higher temperatures, enabling calcined bone to be protected from contamination [54, 59, 65, 66]. Because of this, calcined bone is recognized to be the most reliable source of inorganic C14 for radiocarbon dating, as the elevated crystallinity that accompanies heat alteration at high temperatures protects the Type A and B carbonate substitutions and secondary carbonate incorporated from the burning atmosphere from further alteration, which subsequently can be used to date the burning event [65, 67].

Questions about the changing vulnerabilities of differentially burnt bone prompted our investigations into the characterization of structural changes of modern and archaeological bone burnt at different temperatures. Of specific interest to this study is the timing of the organic loss, and therefore loss of bioapatite protection, in reference to the increases in crystallinity and crystal sizes of bone mineral. Archaeological bone, both unburnt and burnt, is considered in this study as an actualistic reference to relate implications to commonly used zooarchaeological scales of burning intensity, and to monitor the extent of alterations related to the spontaneous postmortem recrystallization of bone mineral which occurs over time in burial environments.

## Materials and methods

### Modern bone sample collection and preparation

A controlled experimental reference collection was created with modern bone to investigate the timing and impact of thermal alteration on bone organic loss, recrystallization indices, and

crystallite size growth. Two horse metacarpals from different individuals were obtained from horses donated to scientific study at the UC Davis School of Veterinary Medicine. Horses were not euthanized in relation to this or any other study, but were humanely euthanized after a poor prognosis of health following extreme cervical osteoarthritis or a femur fracture. Protocol for this procedure was approved by the American Veterinary Medical Association at the UC Davis Center for Equine Health, and was done by trained veterinary staff with great care to ameliorate animal pain, anxiety, and suffering. Metacarpals used for this study were cleaned with simmering water with the addition of borax. Three cow femurs from different individuals were used in this study and were purchased from a local butcher, Adam's Meat Shop (Folsom, CA). These femurs were never frozen, and flesh was scraped manually to prepare for drilling. A diamond drill coring bit was used to produce solid plugs of cortical bone 3mm x 3mm, with weights ranging from 53.2 to 58.1 mg. Coring was constrained to the cortical bone tissue from the mid-diaphysis of both cow and horse bones. Solid bone plugs were specifically utilized in lieu of bone powder to avoid the effects of powder heating, as powder has an increased surface area and would be more reactive to thermal alteration. To fit the dimensions of crucibles used for thermal analyses, plugs were filed with diamond files. Post-experimental heating, three samples were selected for imaging with Scanning Electron (SE) microscopy for visualization purposes. All bone samples were then powdered with an agate mortar and pestle and sieved with 234 µm mesh.

## Modern bone thermal analysis

The controlled annealing of modern bone samples was performed with Setaram Labsys Evo thermal analyzer. Bone core samples were placed in a 100 µl $Al_2O_3$ crucible and air flow 40 ml/min was established. The samples were brought to desired temperatures from 100 to 1200˚C in 100˚C increments, with heating rate 20˚C /min and held isothermally for 30 minutes. The weight change and heat flow traces were recorded continuously and corrected for the baseline. Additional samples were produced at 300 and 700˚C with ramp 50˚C /min and one hour dwell time for comparison.

## Archaeological case study sample collection and preparation

Archaeological unburnt and burnt bone samples were collected from the site of Tolbor-17 (T-17), an open-air locale on a low altitude pass on the western flank of the Khangai Mountains of Northern Mongolia. The Ikh-Tolborin-Gol is part of the Selenga drainage system, the main river feeding Lake Baikal (Fig 1) [68]. This river valley preserves a wealth of Upper Paleolithic (UP) locales including Tolbor-4, Tolbor-15, Tolbor-16, and Tolbor-21[68–71]. Most of the sites document periodic human o ccupations starting with the Initial Upper Paleolithic, ca. 45 ka, until the Holocene. The latter has recently been dated with polymineral post-IR IRSL, Quartz OSL, and radiocarbon to 42.5–45.6 ka, establishing the timing for a movement of population between the Siberian Altai and Northwestern China, contemporaneous with the earliest *Homo sapiens* fossils in the region [68]. The following occupations in the valley are most likely associated with *Homo sapiens*, and are characterized as Upper Paleolithic (UP) in the broad sense. Although it is often assumed that fire is part of the modern human behavioral repertoire allowing expansion into cold climates, evidence of the use of fire in the UP Tolbor locales is rare and has been only briefly reported [72, 73].

Tolbor-17 provides a rare opportunity to investigate faunal remains, as organic material is usually poorly preserved in the region and burnt fauna has not yet been described in detail. Like most of the other locales, T-17 is an open-air environment with a fluctuating low energy run-off, constituting a fluctuating recharge water regime [24, 44]. Initially excavated as a series

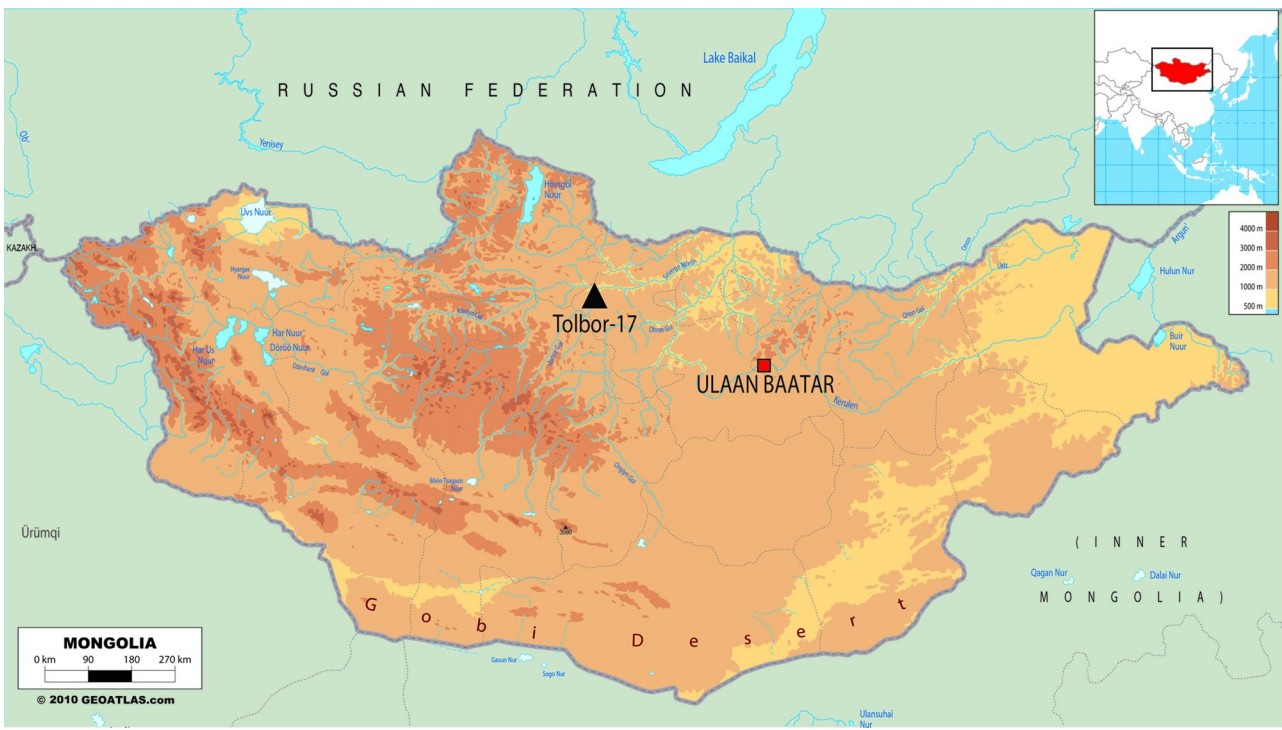

**Fig 1. Map of Mongolia with geographic position of Tolbor-17. Map modified after Geo-atlas.**

of two test pits with dimensions 2 m x 1 m, the excavators at T-17 piece plotted all finds > 2 cm, and the remaining sediment from each bucket volume of excavated material was dry sieved with 4 mm and 2 mm mesh screens, with all material subsequently sorted. The T-17 lithological Unit 3 is characterized by the presence of UP lithic artifacts and organic faunal preservation, despite sedimentary evidence for episodic sheet erosion, prolonged groundwater interaction, chemical weathering, and long surface exposure. Based on its geological setting, the material studied here belongs to the second half of the Marine Isotope Stage (MIS) 3, ca. 40–30 ka cal. BP and is described as UP. Unburnt and burnt fauna have been successfully recovered from Unit 3; however, this assemblage is extremely fragmentary and traditional zooarchaeological analyses based on taxonomic identification and prey selection and processing are still in the preliminary stages.

Mapped (> 2 cm) and screened (< 2 cm-2 mm) faunal remains from the T-17 Unit 3 UP assemblage were cleaned, sorted following the Stiner et al. [27] seven stage visual scale of burning intensity, and weighed (Table 2). No burning was noted in bone > 2 cm except for a single fragment Stiner et al. [27] stage 2, but fauna < 2 cm- 2mm is found to span all stages of burning intensity within Unit 3 of the exposed excavation surface (Table 2). Excluding the relatively large stage 2 fragment, all other indication of thermal alteration from the faunal assemblage was recovered in the screened materials < 2cm, emphasizing the methodological importance of screening for the recovery and recognition of burning, observations noted in previous studies [27, 74].

Of the burnt fauna, a large percentage is nearly or fully calcined, a notable observation due to the recognized mechanical fragility of calcined bone and the unprotected open-air environment of T-17. All excavated fauna was assigned burning stages following Stiner et al. [27] and 20 bones from the same test pit of the Unit 3 assemblage were sub-sampled. A minimum of

**Table 2. T17 Unit 3 fauna burning summary.**

| | Screened (< 2 cm) | Piece plotted (>2 cm) | Total |
|---|---|---|---|
| Burning Stage* | Weight (g) | Weight (g) | Weight (g) |
| 0 | 141.36 | 213.31[†] | 354.47 |
| 1 | 1.04 | 0 | 1.04 |
| 2 | 5.17 | 1.01[‡] | 6.18 |
| 3 | 0.69 | 0 | 0.69 |
| 4 | 1.95 | 0 | 1.95 |
| 5 | 1.37 | 0 | 1.37 |
| 6 | 4.5 | 0 | 4.5 |

*Burning stages following Stiner et al. [27].

[†] n = 81

[‡] n = 1

one category representing bones from this sample were selected for subsequent spectroscopic analyses to confirm heat alteration and investigate organic composition, crystallinity, and crystallite size. Bones were cleaned with ionic water sonication, and two samples representing before and after calcination were selected for imaging with SE microscopy prior to all samples being powdered with a diamond file and an agate mortar and pestle. All archaeological bone powder samples were then sieved with 234 μm mesh. No permits were required for the described study, which complied with all relevant regulations.

## Infrared spectroscopy data collection and analysis

FTIR spectroscopy is a semi-quantitative method which characterizes bond vibrations, absorbed at specific wavelengths of transmitted incident light from infrared radiation, to identify compositional and structural properties of materials [54, 75]. When applied to bone, FTIR spectroscopy can yield valuable information regarding the presence and quality of preserved organic components, as well as the relative degree of structural order, size, and strain of bioapatite crystals [76–80]. This is particularly useful for the detection of organic preservation in samples screened prior to radiocarbon and stable isotope studies, and for diagenetic studies evaluating the integrity of bone mineral [79, 80].

FTIR spectroscopy has additionally had success identifying thermally altered bone, as changes to bone composition and bioapatite crystallinity can be monitored though several heat induced peak transformations which cannot be mistaken for macroscopic staining or bleaching [4, 33, 53–55, 81, 82]. The identification of FTIR spectral peaks associated with the thermal alteration of bone has been extensively documented, with major alterations monitored through: (1) the ratio of carbonate to phosphate present in the sample, the C/P ratio, (2) the depletion of the presence of amide I and II functional groups, representing the organic components of bone, and (3) the presence of heat specific peak splitting, such as the loss of the peak at 874 cm$^{-1}$ correlated to $CO_3^{2-}$ v2 at temperatures over 1000°C, and the PHT shoulder peak at temperatures over 700°C [54, 55, 58, 83–85]. Measures of the crystallinity of a sample can be inferred from the infrared splitting factor (IRSF), which extrapolates the changing size and order of bioapatite crystals through increase of splitting seen in the $PO_4^{3-}$ v4 peaks [54, 58, 83].

Specific peaks relevant to this study and their inferred functional groups include the 1650 cm$^{-1}$ and 1550 cm$^{-1}$ peaks for the measurement of amide I and II, the 874 cm$^{-1}$ and 1415 cm$^{-1}$ peaks indicating presence of the v2 and v3 of carbonate, and the 900–1200 cm$^{-1}$ and 50–600 cm$^{-1}$ spectral regions related to the v3 and v4 phosphate components (Table 3). Additionally,

the appearance of a 625cm$^{-1}$ shoulder peak is attributed here to $PO_4^{3-}$ v4 bending, known as the phosphate high temperature (PHT) [58].

A Nicolet 6700 Fourier transform infrared spectrometer with an ATR attachment and a deuterated triglycine sulfate (DTGS) detector and single bounce diamond crystal was used. The ATR method uses an attachment with a diamond or zinc crystal to produce spectra through the phenomenon of internal reflectance [86–88]. The application of ATR minimizes sample preparation, which in turn minimizes contamination [75, 84, 86]. Spectra were collected with 256 scans in the 4000–400 cm$^{-1}$ frequency region and with an 8 mm spectral range. Each archaeological and modern bone powder sample was retested for quality control.

Eight peak measurements were monitored for 168 scans representing 84 individual samples for this study, 62 modern and 22 archaeological. Each sample was tested twice, and measurements presented here represent the average values of both scans. FTIR-ATR spectra were processed with OMNIC software.

The IRSF measurements were procured for all samples following Weiner and Bar-Yosef [83].

$$\text{Infrared Splittng Factor}: \frac{(565 \text{ cm}^{-1} \; peak \; ht + 605 \text{ cm}^{-1} \; peak \; ht)}{595 \text{ cm}^{-1} \; peak \; ht}$$

An additional measure of the carbonate to phosphate content, the C/P ratio, was also determined for all samples. The C/P ratio decreases with burning and utilizes the 1035 cm$^{-1}$ phosphate peak unaffected by IRSF changes [75, 84].

$$\frac{\text{C}}{\text{P}} \text{ ratio}: \frac{1415 \text{ cm}^{-1} \; peak \; ht}{1035 \text{ cm}^{-1} \; peak \; ht}$$

Other peaks observed for this analysis were noted as they are related to the loss of organics and specific heat-induced changes [54, 55, 58] (Table 3).

## X-ray diffraction

X-ray Diffraction (XRD) can be used to measure the relative sizes of bioapatite crystals [32, 54, 59, 60]. Powder XRD patterns here were obtained using Bruker D2 Phaser and Bruker D8 advance diffractometers using CuKα radiation. Bone powder samples taken from the solid bone plugs and archaeological fauna were spread with ethanol on a zero background silicon sample holder, and run from 10 to 90°2θ with 0.02° step. Dwell time was chosen to obtain at least thousand counts on the most intense peaks. The average crystallite size of analyzed samples was obtained from diffraction peaks broadening using whole pattern fitting (Rietveld refinement) procedure as implemented in Jade MDI software [89]. Diffraction profile was modeled using hydroxyapatite $Ca_5(PO_4)_3OH$ structure (space group P63/m) and pseudo-Voigt profile shape function. The instrumental broadening was accounted for by calibration

**Table 3. FTIR-ATR wavenumbers associated with likely functional groups relevant to this study and the thermal alteration of bone.**

| Wavenumber | Inferred peak assignment | Peak transformation relevant to this study |
|---|---|---|
| 1630–1660 cm$^{-1}$ | organic tissue and water, amide I + II | decrease and absence |
| 1400–1550 cm$^{-1}$ | $CO_3^{2-}$ v3 | 1415 cm$^{-1}$ peak a component of C/P ratio |
| 1028–1100 cm$^{-1}$ | $PO_4^{3-}$ v3 | 1035 cm$^{-1}$ peak a component of the C/P ratio |
| 874 cm$^{-1}$ | $CO_3^{2-}$ v2 | peak loss |
| 565 cm$^{-1}$, 605 cm$^{-1}$ | $PO_4^{3-}$ v4 | growth of 565 cm$^{-1}$ and 605 cm$^{-1}$ and decrease of the 595 cm$^{-1}$ trough utilized for the infrared splitting factor (IRSF); phosphate high temperature (PHT) shoulder growth at 625 cm$^{-1}$ |

with NIST LaB$_6$ profile shape standard. The uncertainties in crystallite sizes are reported as obtained from least squares refinement.

# Results

## FTIR modern samples

The FTIR-ATR spectra were obtained for each sample, 25–1200˚C. All modern samples above 200˚C were found to exhibit spectra indicative of the thermal alteration of bone in oxygen atmospheres supported by previous research (Figs 2 and 3), including the decrease of C/P ratio, decrease of organic components by 300˚C with complete absence seen by 400˚C, the absence the 874 cm$^{-1}$ peak above 1000˚C, and the presence of the PHT peak splitting above 700˚C (S1 and S2 Tables). No differences were indicated in the reheated or increased rate samples taken to 300 and 700˚C from the single-heated or controlled rate counterparts.

The IRSF of all modern samples also followed reported trends in bioapatite crystallinity, with order, size and strain increasing alongside intensifying temperatures and clearly demonstrated with the presence of calcination (Figs 2–4; S1 and S2 Tables) [54, 57]. This increase in crystallinity is seen until 1000˚C, after which there is a marked decrease in IRSF coinciding with the equiaxing of bioapatite crystals (Fig 4). Despite the acceptance of the IRSF metric and general consensus with previously described trends, the values of modern IRSF here do exhibit large variations [54, 57]. This is seen most dramatically in the range of IRSF values reported for all samples at 900˚C in this study (Fig 4; S2 Table). No changes in crystallinity were detected in samples which were reheated or heated with increased rates.

## FTIR archaeological samples

The FTIR-ATR spectra produced from the T-17 archaeological collection is consistent with the Stiner et al. [27] stages of temperature intensity assignments based on color alteration. Unburnt bone is supported as non-heat altered and burnt bone does not indicate signs of intrusive staining or bleaching (Figs 5 and 6). Good agreement is seen with the relative decreases of C/P ratio and the loss of organic components by Stage 3 (fully carbonized) between the archaeological and modern samples (Figs 5 and 6; S3 and S4 Tables). The appearance of the PHT with bones identified as Stage 5 supports the presence of temperatures above 700˚C at T-17, although the continued presence of $CO_3^{2-}$ v2 inferred by the 874 cm$^{-1}$ peak indicates temperatures likely did not reach above 1000˚C (Figs 5 and 6; S3 Table).

The IRSF of the T-17 samples also follows the trends of the experimental modern collection, with gradual increases seen through Stage 3 (fully carbonized) and demonstrably higher values reported with the presence of calcination at Stages 4, 5, and 6 (Fig 4; S3 and S4 Tables). Elevated values are not seen within the Stage 0 unburnt samples of T-17 bone, demonstrating that the spontaneous recrystallization of bone mineral that accompanies diagenesis does not exceed here values of modern or archaeological samples which have been burnt at low or high temperatures. As expected, the IRSF values can distinguish between calcined and non-calcined samples but cannot distinguish between low temperature burning samples.

## XRD modern samples

The results of the XRD analyses on the modern samples demonstrate the increasing crystallite size correlated to temperature, specifically above temperatures of calcination (700˚C) (Fig 7; S2 Table). An average size threshold is clearly noted, with all samples unburnt through 600˚C averaging 9 nm, while all samples burnt at 700˚C jump to an average of 41 nm (Fig 7). An additional increase in crystallite size by approximately 30 nm is noted at 900˚C, coinciding

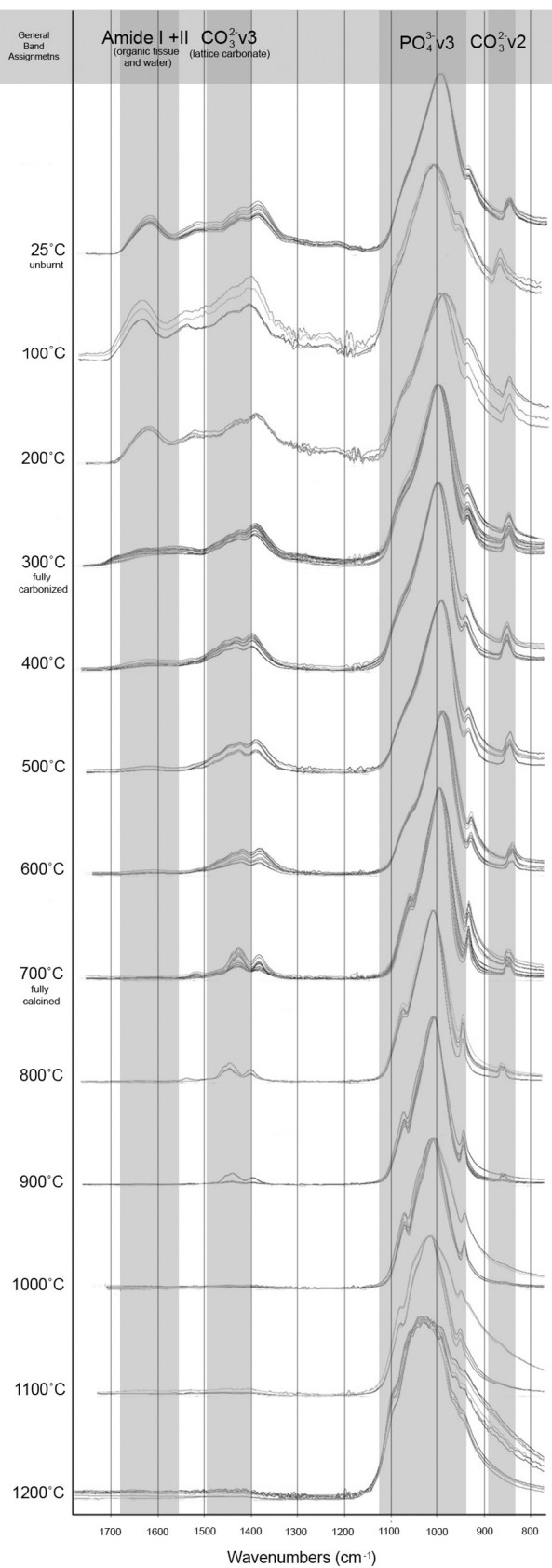

**Fig 2. FTIR-ATR spectra of experimentally modern burnt samples grouped by with functional groups highlighted in the range of 1700–800 cm[-1].**

with the fusion stage of thermal alteration of bone, with samples reaching an average of 72 nm (Fig 7). The XRD results also demonstrate that within fully calcined samples, corresponding to Stiner et al. [27] burning scale Stage 6, two temperature thresholds can be distinguished: the maximum crystal size of calcined bone >700°C not exceeding 70 nm, and the maximum crystal size of calcined bone >900–1200° C exceeding 90 nm (Fig 7). This study therefore suggests average crystallite size as a metric to distinguish temperatures obtained within calcined bone classified as Stiner et al. Stage 6, > 700°C and > 900°C.

## XRD archaeological samples

The average crystallite size measurements for the archaeological T-17 bioapatite samples confirm that the natural postmortem recrystallization of unburnt archaeological bone does not exceed crystal sizes of modern or archaeological bones burnt to low temperatures. The crystallite sizes do increase considerably with the presence of calcination at Stiner et al. [27] Stages 4, 5, and 6 and can be distinguished from lower temperature burning and unburnt bone (Fig 7; S4 Table). Utilizing the metric of crystal size, we also suggest that for two of the archaeological samples of Stage 6 fully calcined bone there is evidence of burning > 900°C—<1100°C.

## Discussion

The measurements presented here regarding the loss of organics, the degree of crystallinity, and the crystallite sizes are in consensus with trends reported by previous studies, with small variance likely introduced by the burning of solid bone versus powdered samples and different experimental sample preparation and burning regimes. Our results demonstrate that for bone burnt to lower temperatures before calcination, Stiner et al. [27] Stages 1–3, bioapatite retains the small reactive crystal sizes of unburnt bone but faces rapid organic loss. The depletion of the organic component is most complete just prior to temperatures of calcination (~700°C). This disintegration of the collagen in charred or carbonized bone is recognized to produce a very open porosity, leaving bone mineral burnt highly exposed [59].

The increase in crystallinity and crystal size growth seen by 700°C corresponding to calcination and Stiner et al., [27] Stages 4–6 immediately reduces the surface to mass ratio and active surface area of bone mineral. This lower surface area results in a product with a lower solubility potential than unburnt and charred or carbonized bone. For fully calcined Stage 6 bones which have also been exposed to temperatures above 900°C in oxygen atmospheres, there is the additional benefit of reported compaction and closing of the porosity, further limiting the access of any destructive agents to the exposed crystallites [59]. The heat induced dimensional changes are also considerably larger than the small amounts of diagenetic recrystallization which assists with the prevention of contamination in fresh bone, as evidenced by the inclusion of unburnt archaeological samples.

Diagenetic agents such as water dissolution are therefore inferred here to be a serious threat to bone burnt to temperatures under 700°C, because partially and wholly carbonized and charred burnt bone has low levels of crystallinity and small crystals which preserve a reactive surface area similar to unburnt fresh bone at the time of deposition. The additional lack of organic protection, open porosity, and likely small fragment sizes due to the recognized friability of burnt bones calls attention to the further vulnerability of charred and carbonized fauna in Stiner et al. [27] Stages 1–3 due to bioapatite dissolution and disintegration.

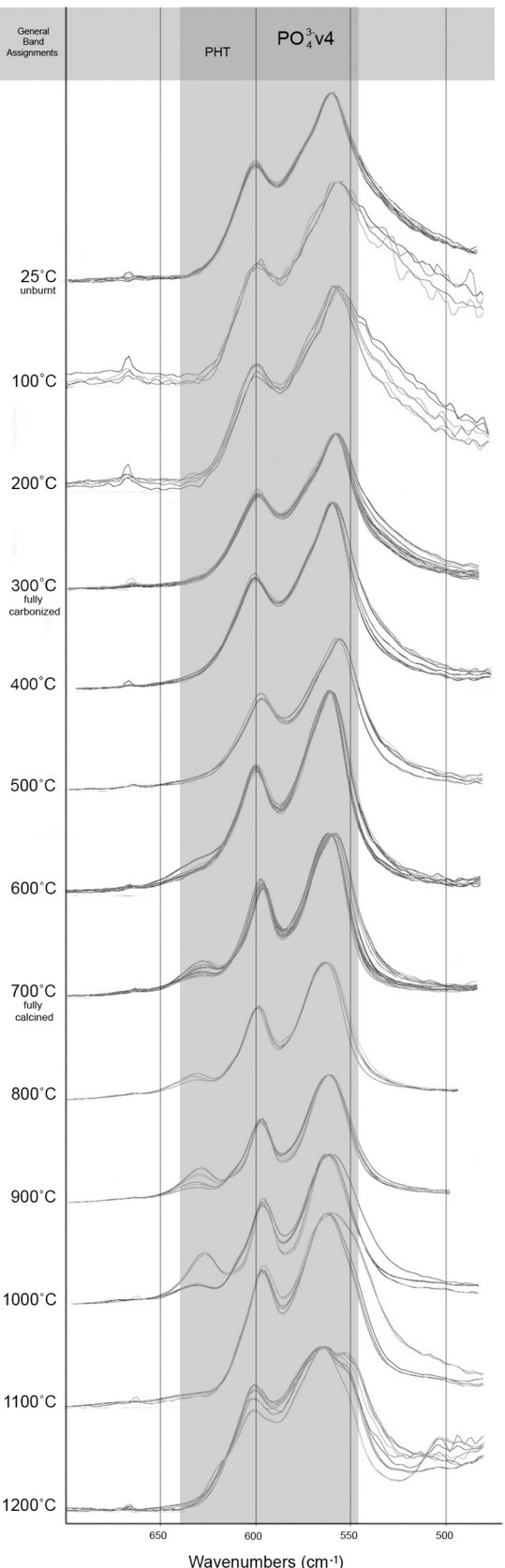

**Fig 3. FTIR-ATR spectra of experimentally modern burnt samples with functional groups highlighted in the range of 700–500 cm$^{-1}$.**

Correspondingly, the lack of organic presence in burnt bones, especially bones corresponding to Stiner et al. [27] Stages 3–6, is interpreted here to be a less attractive target for the diagenetic agent of microbial action than fresh and fleshed unburnt bone.

The results of this study propose a hypothesis of differential survivorship in bones burnt to different temperature thresholds in assemblages of archaeological fauna with fluctuating water movement. This study would further suggest that bones with low density and high porosity, such as trabecular and juvenile bone, are further susceptible to dissolution if charred or carbonized. This is due to the assumption that open porosities, likely small fragment sizes, and no organic protection exacerbates the access of diagenetic agents such as fluctuating water to their small and reactive crystals.

Bone which has been calcined, including trabecular and juvenile bone, potentially has a greater likelihood to resist dissolution due to the large crystal sizes and greater thermodynamic stability. These properties have been similarly recognized to contribute to the suitability of calcined bone for inorganic C14 dating analyses [65]. Calcined bone has been shown to be more mechanically fragile than both unburnt and charred or carbonized bone, however, and traumatic action including post-depositional movement and trampling can be highly destructive. Calcined bone therefore may be expected to preserve well in a variety of depositional environments, albeit in small fragment sizes recoverable only through screening [27].

The recognition of ancient fire in the archaeological record remains a major challenge, including discerning the presence of fire with anthropogenic origins, and the utilizing the properties of burned materials to inform on specific human behaviors of interest. Current standards for identifying anthropogenic fire considers multiple techniques and lines of evidence, including the study of small, fragmented fauna and investigations at the microscopic scale [90]. This microcontextual approach is especially necessary in contexts when connecting fire to hominin behavior is unknown or unclear [90]. Hypotheses regarding the ubiquity of fire use by hominins is currently under investigation, with habitual fire use by Neanderthals during cold and arid climatic episodes being called into question by some [91], and others noting that the energetic costs of producing and maintaining fire might be substantial in an environment where fuel is lacking [92, 93]. Even though fire is largely expected to be an integral

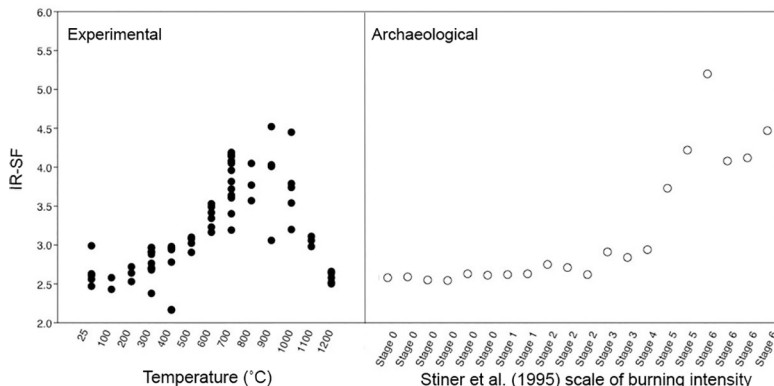

**Fig 4. Infrared Splitting Factor (IRSF) of the experimental modern and archaeological collection measured from FTIR-ATR spectra following Weiner and Bar Yosef [81].**

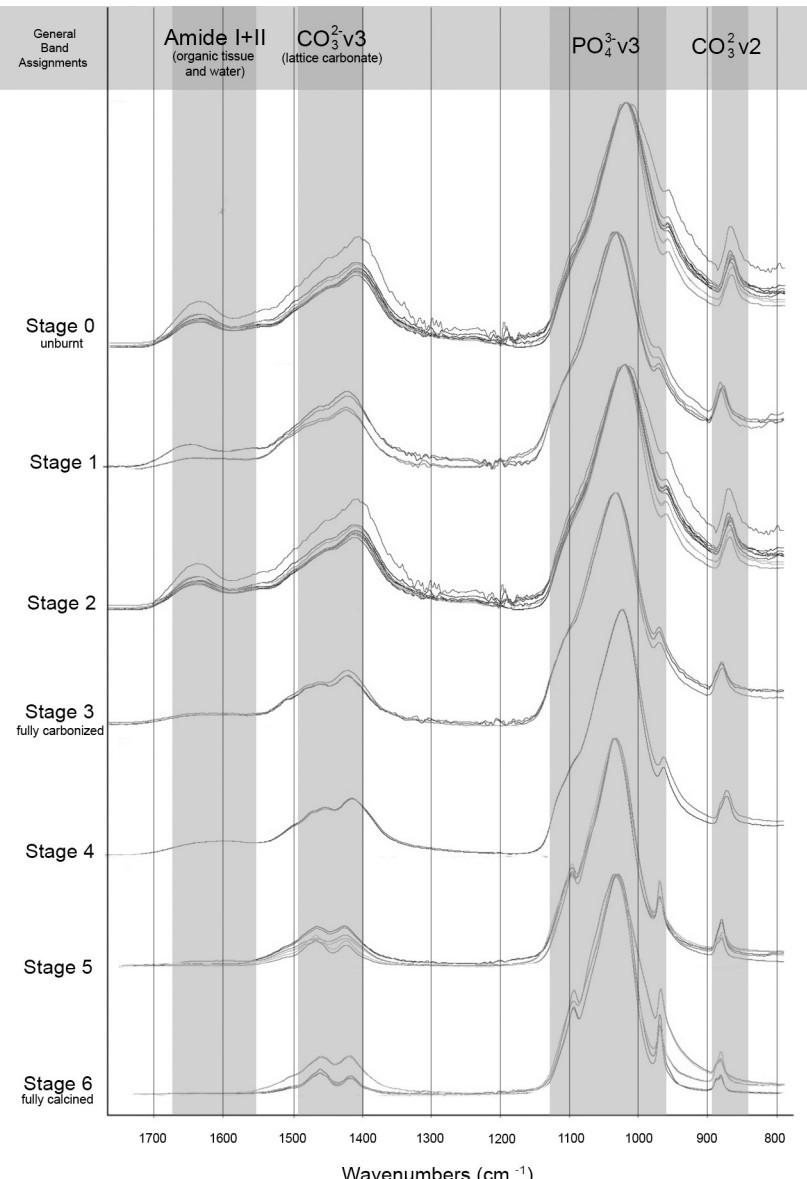

**Fig 5. FTIR-ATR spectra of archaeological fauna from T-17 grouped by stage of burning intensity following Stiner et al. [27] with functional groups highlighted in the range of 1700–800 cm⁻¹.**

component of *Homo sapiens* technology, we note that Pleistocene hunter gather groups in the Ikh-Tolborin-Gol probably faced comparable economic constraints in continental climate.

In the Ikh-Tolborin-Gol, a wealth of Upper Paleolithic sites documents episodic human occupations from the earliest movements of *Homo sapiens* across the Eurasian steppe until the Holocene [68, 73]. Yet, direct evidence for the use of fire is rare and poorly preserved. At Tolbor-15, reddish-brown sediment and apparent accumulation of ashes have been described as 'hearths'. The Archeological Horizons (AH) 6 and 7 are dated ca. 40.5–37.4 ka cal BP, the use of fire has been presented as an adaptive behavior to cold and arid climate [94, 95]. At Tolbor-21, concentrations of discolored sediments and rocks have been reported within AH4 and dated of ca. 42.5–41.5 ka cal BP but a formal identification of combustion features could not

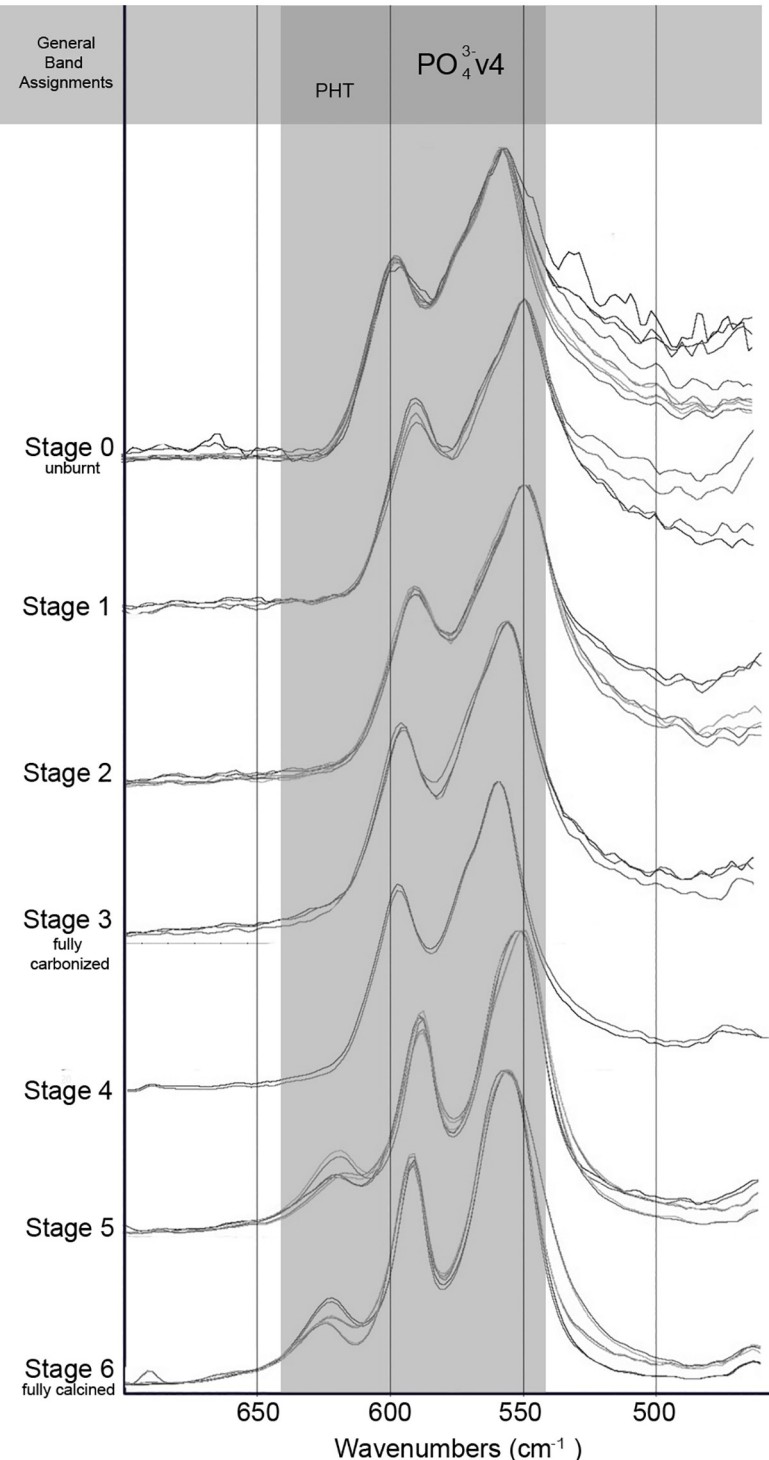

**Fig 6. FTIR-ATR spectra of archaeological fauna from T-17 grouped by stage of burning intensity following Stiner et al. [27] with functional groups highlighted in the range of 700–500 cm⁻¹.**

be confirmed [96]. At the neighboring site of Tolbor-16, two potential combustion features have been identified in Pit 1 within AH5 and AH6, dated of ca. 38.5–37.2 ka and ca. 45.5–44.5ka cal BP respectively [69]. With a singular burnt bone found within the discolored area

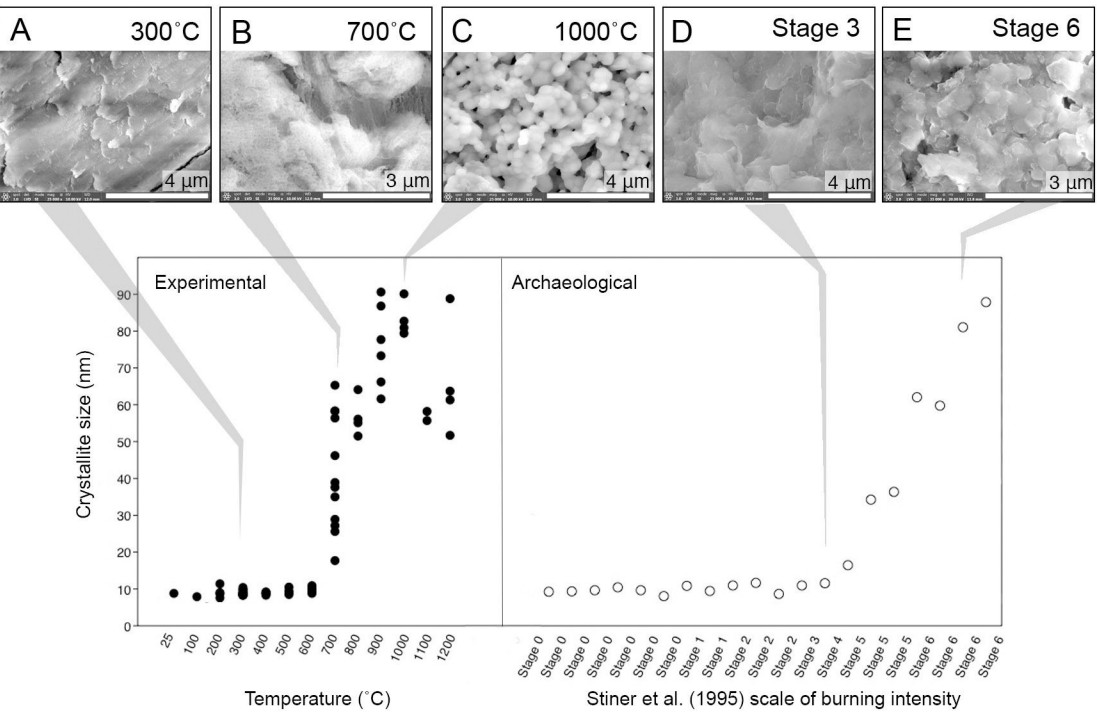

**Fig 7.** XRD results of averaged crystallite size (nm) alongside selected SE images of experimental modern (A-C) and archaeological (D-E) bone to visualize the changes to crystal shape with heat alteration. Images A and D highlight the small, platelike shape of bioapatite burned to at least 300˚C. Images B and C, both modern, demonstrate morphological range of bones burnt above 700˚C, both of which are fully calcined and are considered Stage 6 on the Stiner et al. [27] scale of burning intensity. The initial growth of hexagonal crystals ~700˚C (B) and eventual equiaxing and further growth of crystals above 900˚C (C) are clearly seen in images B and C, and when compared to the archaeological sample of a fully calcined Stage 6 bone (E), the bioapatite crystal morphology corresponds to the size and shape of experimental bones burned above 900˚C. Magnification and instrument details described in S1 Appendix.

and a few isolated charcoals found outside of this perimeter at Tolbor-16, the identification of anthropogenic fire is yet to be confirmed [68, 69].

The potential combustion features described above have in common to be directly associated with archaeological material in a context where preservation of bones is poor. They also share a lack of charcoal associated with the discolored sediment. Overall, there is little evidence for anthropogenic fire during the Pleistocene of the valley. Whether this pattern reflects human behavior, preservation, or visibility bias should be clarified to address the evolutionary significance of this technology in the valley. For example, the spectroscopic analyses presented here includes temperatures ~700–900˚C that are consistent with a high intensity fire, potentially anthropogenic. Given the small number of such bones, additional sampling micromorphological investigations and spatial analyses should confirm such interpretation. Furthermore, our results also show that at T-17 there was very little representation of bones heated to lower temperatures, all within a range that would affect their preservation given contact with water. Given that evidence of active and fluctuating water movement are observed in most of the excavated sites in the Ikh-Tolborin-Gol, we consider that the hypothesis of differential survivorship is worth testing with a full zooarchaeological profile of T-17 Unit 3.

To summarize, small fragments of both burnt and unburnt bone are preserved at T-17, which is exceptional for the region. It is therefore a key site to address issues of fire presence and preservation. The samples from T-17 included in this study both demonstrate the applicability of the observed structural changes in an archaeological assemblage of burnt bone, but

also represent a first step toward the study of fire technology at the microscopic scale in the Ikh-Tolborin-Gol. Archaeological implications are that in the Ikh-Tolborin-Gol the lack of evidence for the use of fire during the Pleistocene may not only reflect human behavior. Instead, it could also be the result of a preservation bias due to complex combination of factors such as the temperatures to which bones were heated and the contact with circulating water. Testing this hypothesis should lead to a comprehensive and contextualized study of the nature and properties of the fire at T-17, and its relevance to Upper Paleolithic technological adaptations in the Tolbor valley.

## Conclusion

The spectroscopic measurements reported here encompass a large reference collection of burning in highly controlled conditions 100–1200˚C in fresh solid bone cores, extending previous modern and archaeological datasets that consider the structural and compositional changes of burnt bone. Especially of note is the average crystallite size difference between bone burnt above 700˚C, a metric which may be used to distinguish greater burning temperatures within fully calcined archaeological bone, which are all considered as Stiner et al. [27] Stage 6 based on visual coloration.

Our results highlight the vulnerability of charred and carbonized bone to diagenetic agents and generate hypotheses regarding the differential survivorship of bone burnt to different temperatures. Here the nature of the hydrological environment is proposed to be a significant threat to bone burnt to low temperatures, as water fluctuation is a large factor in fresh bone bioapatite dissolution and charred and carbonized bone has similar bone mineral properties but has had organic protection already eliminated prior to burial.

This study reiterates the importance of small bone fragments, as excluding the screened faunal material from T-17 would have obscured the presence of fire almost entirely. The identification and recognition of further biases in addition to mechanical fragility of burnt bone, and the variation within bones burnt to different temperatures, is emphasized here as burnt bone exhibits a range of structural and thermodynamic properties across zooarchaeological scales of visible burning intensity.

While burnt bone can provide valuable and detailed information on ancient fires relevant to human behaviors, this study demonstrates that the structural properties and vulnerabilities of burnt bone vary widely between burning temperature thresholds, which may result in biased differential survivorship. This has large implications for studies utilizing the properties, presence, and distribution of temperatures of burnt bone to reconstruct the visibility and intensities of ancient fire, especially low temperature fire events.

## Supporting information

**S1 Appendix. SE microscopy imaging instrument and magnification details.**
(PDF)

**S1 Table. Experimental modern bone FTIR-ATR relevant peak height values.**
(PDF)

**S2 Table. Calculated experimental modern sample values of FTIR C/P, IRSF, and bioapatite crystal size average as measured from XRD.**
(PDF)

**S3 Table. FTIR-ATR Unit 3 archaeological sample relevant peak height values.**
(PDF)

**S4 Table. Calculated archaeological T-17 Unit 3 C/P and IRSF values, and bioapatite crystal size average as measured from XRD.**
(PDF)

## Acknowledgments

The authors would like to thank Dr. Teresa Steele for her continued feedback and illuminating discussions, and Cody Prang and Chris Gallo for their many helpful explanations. We additionally thank Drs. Eerkens, Darwent, and Parikh for their generosity and lab resources, as well as the patience, assistance, and expertise of the Peter A. Rock Thermochemistry Laboratory, NEAT research group, and the Parikh Environmental Soil Chemistry Group at the University California Davis. The authors also recognize the support given from the UC Davis Veterinary Medical Teaching Hospital, the UC Davis JD Wheat Veterinary Orthopedic Laboratory, the UC Davis AMCaT Laboratory, the UC Davis DHI Transdisciplinary Research Cluster "The Cluster for Archaeology and Soil Synergy", the Prehistory department of Université Liège, and the Center for Experimental Archaeology at UC Davis (CEAD).

## Author Contributions

**Conceptualization:** Giulia Gallo, Matthew Fyhrie.

**Data curation:** Giulia Gallo, Matthew Fyhrie, Sergey V. Ushakov.

**Formal analysis:** Giulia Gallo, Matthew Fyhrie, Sergey V. Ushakov.

**Funding acquisition:** Giulia Gallo, Masami Izuho, Nicolas Zwyns.

**Investigation:** Giulia Gallo, Matthew Fyhrie, Sergey V. Ushakov.

**Methodology:** Giulia Gallo, Matthew Fyhrie, Sergey V. Ushakov, Alexandra Navrotsky.

**Project administration:** Nicolas Zwyns, Alexandra Navrotsky.

**Resources:** Masami Izuho, Nicolas Zwyns, Alexandra Navrotsky.

**Supervision:** Cleantha Paine, Sergey V. Ushakov, Masami Izuho, Byambaa Gunchinsuren, Nicolas Zwyns, Alexandra Navrotsky.

**Visualization:** Giulia Gallo.

**Writing – original draft:** Giulia Gallo, Nicolas Zwyns.

**Writing – review & editing:** Cleantha Paine, Sergey V. Ushakov, Masami Izuho, Nicolas Zwyns, Alexandra Navrotsky.

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
