## [Decision Letter · Decision Letter 0]

14 Aug 2020

PONE-D-20-19845

Characterization of structural changes in modern and archaeological burnt bone: Implications for differential preservation bias

PLOS ONE

Dear Dr. Gallo,

Thank you for submitting your manuscript to PLOS ONE. After careful consideration, we feel that it has merit but does not fully meet PLOS ONE’s publication criteria as it currently stands. Therefore, we invite you to submit a revised version of the manuscript that addresses the points raised during the review process.

We look forward to receiving your revised manuscript.

Kind regards,

Justin W. Adams, Ph.D.

Academic Editor

PLOS ONE

Journal Requirements:

2. In your manuscript, please provide additional information regarding the specimens used in your study. Ensure that you have reported specimen numbers and complete repository information, including museum name and geographic location.

For more information on PLOS ONE's requirements for paleontology and archaeology research, see https://journals.plos.org/plosone/s/submission-guidelines#loc-paleontology-and-archaeology-research.

Additional Editor Comments (if provided):

Thank you for your submission.

In reviewing the feedback from all three Reviewers, several major factors have been highlighted that will need to be considered with the manuscript in its current form. Setting aside some of the suggestions to provide greater literature integration, restructure of the manuscript to improve clarity, and potentially to reduce the background content into supplementary materials to improve readability, the primary concerns raised by the reviewers focus on: 1) providing sufficient supporting data (particularly SEM data) to support the argument and interpretations made within the manuscript; and 2) greater integration of the prmary study topic (characterisation of burnt bone) with other taphonomic processes that co-vary in assemblages and impact interpretability. I encourage you to read these comments carefully, particularly those from Reviewer 1 and 3 when revising the manuscript for resubmission. I'd also note that I would expect, given the scope of the comments and requests for greater integration and expansion of data presentation, that the resubmission will require another round of peer-review.

Reviewers' comments:

Reviewer's Responses to Questions

**Comments to the Author**

1. Is the manuscript technically sound, and do the data support the conclusions?

Reviewer #1: Yes

Reviewer #2: Yes

Reviewer #3: Partly

2. Has the statistical analysis been performed appropriately and rigorously? 

Reviewer #1: N/A

Reviewer #2: Yes

Reviewer #3: Yes

3. Have the authors made all data underlying the findings in their manuscript fully available?

Reviewer #1: Yes

Reviewer #2: Yes

Reviewer #3: No

4. Is the manuscript presented in an intelligible fashion and written in standard English?

Reviewer #1: No

Reviewer #2: Yes

Reviewer #3: Yes

5. Review Comments to the Author

Reviewer #1: The paper is very interesting and suitable for Plos One, of broader utility. However, the originality of the paper needs to be more explicitly described, crystallinity changes are referred to burning but very little if any is referred to diagenetic influence in burnt archaeological specimens (which is the objective of the paper but finally not clearly discussed). A reorganization of the text and more clear statements and aspects to be discussed to demonstrate the goals of the paper is needed.

All referred to burnt effects from modern collection applied to fossils is well and clearly described. I believe that the inference of all analyses is that bones below carbonization are underrepresented in the fossil site T-17 unit 3. I believe the problem is to better describe what the Unit 3 of T17 is characterized for. One apparent conclusion is that burning is the cause of fragmentation, but this is not true. Burning alone does not cause fragmentation, a subsequent movement or effort may produce fragmentation, but indication of which movement or effort acted after burning is not described. Further, fragmentation before and after burning can be caused by a large number of taphonomic agents (e.g. butchery, trampling, weathering, corrosion...). None of them are described in the paper. Definitively, the paper needs a better and more exhaustive description of the fossil assemblage: number of specimens, size of fossils, surface modifications, how many fossils are in each burning stage or which have also been affected by any other taphonomic agent both before and after burning should be described or included in a table...

All should be described more in detail, all should be reorganized and described in a more organized way according to objectives and conclusions, right now it is very disorganized, lacks basic information and conclusions are not clear enough, rather an act of faith...and the paper needs several reading to follow the conclusions.

Some suggestions to the authors are included in the file attached.

Reviewer #2: The article "Characterization of structural changes in modern and archaeological burnt bone:

Implications for differential preservation bias" by Gallo et colleagues is certainly interesting, well written and argued and with a correct methodology.

Although the performed methodological procedures are well known, the modern bone analysis and the discussion involving the possible preservation bias is quite interesting.

I think that the article can be published in PlosOne after some minor revisions:

1. I like very much the introductory part, it is a very good compilation/resume of the state of the art of bone transformation research. I will suggest to try to resume all the information in a graph where the X axe is the burning temperature/time (qualitative and/or quantitative) and all the mentioned parameter-changes (Y axe) are depicted.

2. First paragraph of page 13: Describe better the sedimentological/postdepositional features indicating the mentioned features/processes. Any images of the deposits and/or bone remains? They will be very welcome.

3. Include/discuss more FTIR-ATR related references for archaeological studies in the discussion, please reivew this recent work and references therein:

Iriarte, E., García-Tojal, J., Santana, J., Jorge-Villar, S.E., Teira, L.C., Muñiz, J.R., Ibañez, J.J. (2020). Geochemical and spectroscopic approach to the characterization of earliest cremated human bones from the Levant (PPNB of Kharaysin, Jordan). Journal of Achaeological Science: Reports, 30. 102211. doi: https://doi.org/10.1016/j.jasrep.2020.102211.

Reviewer #3: The manuscript by Gallo et al discusses an interesting topic related to the behaviour and diagenesis of archaeological bones after exposition at different T. Authors compare mineralogical data from modern samples and archaeological ones. The results are interesting, but the manuscript needs more work before it can be suitable for publication. I summarise my suggestions in the following bullet points.

- The abstract is not informative about the results but mostly talk about methods. I suggest to rewrite it.

- Introduction: you explain the aims of your study, but I think that a major implication of your work is not considered in this paper. I suggest to discuss in the introduction section (and maybe in the discussion or conclusion) a further implication: are your results relevant in the field of radiocarbon dating on bones or collagen and DNA extraction? I think that you may discuss the importance of changes in crystallinity of bone apatite in the context of radiocarbon dating of archaeological bones.

-The Background section is really too long. It seems a summary of the background chapters of a PhD dissertation. I suggest to strongly reduce this section and possibly to add some parts to supplementary material or add a new table summarising available information about bones, porosity, diagenesis, solubility. All information reported in this part is very interesting but are redundant in a scientific paper.

- Looking to the cited references, I noted that you completely missed the many papers published by Gregorio dal Sasso of the University of Padua (Italy). His PhD project was dedicated to the issue of archaeological bones diagenesis after burying and he investigated in details the behaviour of buried bones in terms of changes in crystallinity and interaction with water in the archaeological deposit. I strongly suggest especially to consider the paper elaborating an universal curve for apatite crystallinity (https://www.nature.com/articles/s41598-018-30642-z).

- In the method section, please explain how you selected samples from the archaeological sequence of Tolbor. Moreover, In the result section I strongly suggest to add information about collected bones. For instance, the description of bones and their preservation is missing. The same for the model bone samples. Also, some pictures would help.

- I would also suggest to add (maybe in supplementary material) the section of Tolbon indicating the stratigraphic position of sampled bones. Please condor also that if bones come from different parts of the archaeological despots, they may have suffered different (I mean local) diagenetic process (e.g. related to water percolation or variations of pH values of the host archaeological layer); can this influence your results? the sole map of the study region is completely useless if you don't supply data on the stratigraphy of the site. Finally, it is not clear why you selected this specific archaeological sequence to carry out your experiments.

- The description of methods is also very long and many informations are very basic. I suggest to shorten this part of the manuscript or move some parts to supplementary material.

- I think the corpus of data described in the result section is robust and it is a good starting point for your discussion. Unfortunately, you do not present and discuss results of SEM investigation. Scanning microscope, I guess, may inform about the modification of apatite crystals of your samples; for instance changes in crystal shape or orientation, or can explain the evolution of diagenesis recording the process of recrystallization (evidence of dissolution and recrystallisation are very evident under the SEM). As you cited SEM in the Methods section, I think you should have interesting data to show. The only SEM images are in Fig. 7, but they are not discussed. SEM data would increase also the quality of discussion.

- Discussion: at the moment the discussion is mostly the comparison between data from modern samples and the archaeological ones. I suggest to expand the discussion also because the interaction between water and buried bones is not very clear. What is the specific potential influence of circulating water on bones heated at different T? As I already said, this may have huge implication intern in radiocarbon dating. Bone apatite is used in dating, but it is not the best, because it often underwent recrystallisation or contamination by CaCO3 (see Cherkinsky and di Lernia, Radiocarbon 2013 and some of the papers by Dal Sasso). Also, water circulating in an archaeological deposit is generally quite rich in dissolved CaCO3 and there are major exchanges of Ca2+ between fluids and bone apatite; you can also have deposition of CaCO3 along with recrystallisation of bone apatite. What is the interaction of Ca-rich fluids and bones heated at different T? You shall consider this fundamental interaction and discuss it on the basis of your data. Two main points are laking in the discussion, but they are mandatory, the interaction between CaCO3 rich water and archaeological bones, and the implication of your study to the field of radiocarbon dating of archaeological bones.

6. PLOS authors have the option to publish the peer review history of their article (what does this mean?). If published, this will include your full peer review and any attached files.

Reviewer #1: **Yes: **Yolanda Fernández-Jalvo

Reviewer #2: No

Reviewer #3: No

---

## [Author Response · Author response to Decision Letter 0]

14 Feb 2021

The following is copied from our Response to Reviewers document uploaded as an attachment.

Response to Reviewers: 

Reviewer 1

…However, the originality of the paper needs to be more explicitly described, crystallinity changes are referred to burning but very little if any is referred to diagenetic influence in burnt archaeological specimens (which is the objective of the paper but finally not clearly discussed). A reorganization of the text and more clear statements and aspects to be discussed to demonstrate the goals of the paper is needed.

Thank you for your comment and for all your helpful suggestions. The manuscript is greatly improved with your feedback. To highlight the originality of the paper we have rewritten major portions of the introduction and background sections, attempting to summarize concisely the exact goals of the study and inclusion of the modern and archaeological samples. We believe this will reframe and clarify the role of the Tolbor-17 material for this study, as it is our intention that the Tobor-17 archaeological burnt bone serve as a comparative reference sample of changes in bioapatite crystallinity sizes and organic content for this study. Future zooarchaeological studies planned for Tolbor-17 will consider the hypothesis of differential preservation closely with supporting faunal and geoarchaeological data. We have therefore modified our conclusions to reflect this restructuring, using our results from the study as hypothesis generating observations for further closer study, with the intention to follow up more substantially in the forthcoming full zooarchaeological study of Tolbor-17.

Burning alone does not cause fragmentation, a subsequent movement or effort may produce fragmentation, but indication of which movement or effort acted after burning is not described. Further, fragmentation before and after burning can be caused by a large number of taphonomic agents (e.g. butchery, trampling, weathering, corrosion…).

Thank you for your observation. This perspective and related information will be the major focus of the next study planned for this project: a full zooarchaeological study of the Tolbor-17 faunal material. This planned study will include traditional zooarchaeological analyses including identifiable specimens, fragmentation degree, surface modifications, and preservation. We seek to describe many anthropogenic behaviors present in the Tolbor-17 fauna in the future study, and will pay specific attention to burning to test the hypothesis of differential preservation suggested by our current manuscript. This zooarchaeological investigation has had initial data collection completed in December 2019, with further analyses planned with international travel restrictions are lifted. 

Definitively, the paper needs a better and more exhaustive description of the fossil assemblage: number of specimens, size of fossils, surface modifications, how many fossils are in each burning stage or which have also been affected by any other taphonomic agent both before and after burning should be described or included in a table.

We believe that through the reframing and clarification of the purpose of the archaeological material (as a reference sample confirming the visibility in changes of bioapatite crystallinity sizes and organic preservation in an archaeological assemblage and relating described changes to zooarchaeological observational scales of burning intensity) will clarify the intention of the archaeological sample inclusion. A sample of text addressing this specifically is now included in the introduction and is highlighted in the manuscript with tracked changes on page 3. We also have reframed our discussion of the archaeological material in the context of hypothesis generation, and the descriptive fossil assemblage desired here will be the focus of the next paper engaging with the planned zooarchaeological study of the full Tolbor-17 fauna from Unit 3. Language addressing this has been rewritten and is highlighted in page 23 of the manuscript with tracked changes.

Reviewer 2

I like very much the introductory part, it is a very good compilation/resume of the state of the art of bone transformation research. I will suggest to try to resume all the information in a graph where the X axe is the burning temperature/time (qualitative and/or quantitative) and all the mentioned parameter-changes (Y axe) are depicted.

Thank you very much for your time and helpful feedback for our manuscript. It was a concern of ours that the diverse researchers who may be interested in this topic (ranging in disciplines from zooarchaeology to material scientists and thermochemists) may need a full background to provide a comprehensive understanding of the many complex systems being considered, so it was our original intention to have a very extensive background section. From the feedback of our other reviewers, however, it was suggested that this be shortened and made more concise. We believe the revisions have keep the scope and scale of the original intention for the background section while eliminating some more extraneous details and clarifying the language. 

Additionally, we have tried to create several versions of a schematic following your suggestion. All versions were found to be unsatisfactory, both due to the changing scenarios around burning with or without oxygen, and the variance found in reported data regarding porosity which we do not directly investigate in our own study. Our solution was to clarify the language regarding the timing of the described mechanisms, primarily found in our revised discussion section is highlighted on page 22 of the manuscript with tracked changes. 

First paragraph of page 13: Describe better the sedimentological/postdepositional features indicating the mentioned features/processes. Any images of the deposits and/or bone remains? 

We thank you for your suggestion. We have described the sedimentological context to the extent of our current knowledge, and have revised our text regarding our intention for the archaeological material to be a reference sample into bioapatite crystallinity and organic components of archaeological burned bone. In our forthcoming study on the full zooarchaeological material from Tolbor-17, of which the hypotheses generated by this paper will be tested with further studies, we plan on including a greater amount of detail on the post-depositional taphonomic processes and the greater geological context of Unit 3. 

Include/discuss more FTIR-ATR related references for archaeological studies in the discussion, please reivew this recent work and references therein: Iriarte et al., 2020

We are grateful for your encouragement to include the very relevant work of Iriarte et al., 2020. We have now included this work in our discussion of the potential of burnt bone to make advances in our understanding of past human behavior.

Reviewer 3

The abstract is not informative about the results but mostly talk about methods. I suggest to rewrite it.

Thank you for your suggestion, and for your constructive feedback. The abstract has been fully rewritten to address your suggestion. 

Introduction: you explain the aims of your study, but I think that a major implication of your work is not considered in this paper. I suggest to discuss in the introduction section (and maybe in the discussion or conclusion) a further implication: are your results relevant in the field of radiocarbon dating on bones or collagen and DNA extraction? I think that you may discuss the importance of changes in crystallinity of bone apatite in the context of radiocarbon dating of archaeological bones.

We thank you for observation, and we have included the discussion of C14 dating of the inorganic component of bone in our revised manuscript. We are grateful for the expansion of the implications and relevance of our study, and this has provided an additional opportunity for us to contextualize and highlight the massive bioapatite mineral structural reorganization of calcined bone. Importantly, it is the same mechanisms which are responsible for the usefulness of calcined bioapatite for C14 dating (thermodynamically stable crystals that can resist contamination) that we describe in this study first at temperatures of calcination (~700˚C), and also a second threshold at higher temperatures (~900˚C). This discussion and many relevant citations is included in our revised section on burnt bone diagenesis which is highlighted on page 10 our manuscript with tracked changes.

We believe our revised text and clarification on the timing of organic loss, and therefore waning usefulness for collagen and DNA extraction, will benefit those audiences as our results provide reference spectroscopic datasets illustrating the presence and quick decay of organic components in burnt bone.

The Background section is really too long. It seems a summary of the background chapters of a PhD dissertation. I suggest to strongly reduce this section and possibly to add some parts to supplementary material or add a new table…

Thank you. We had major concerns about the background necessary for all varied audiences who may be interested in this paper, and it was our original intention to have a detailed background to meet those needs. Considering your suggestion, we have completely rewritten and restructured our background sections. We believe this revised version still has the breadth to inform readers from different disciplines, but is no longer dense and focused on tangential information. 

I noted that you completely missed the many papers published by Gregorio dal Sasso of the University of Padua (Italy). His PhD project was dedicated to the issue of archaeological bones diagenesis after burying and he investigated in details the behaviour of buried bones in terms of changes in crystallinity and interaction with water in the archaeological deposit. I strongly suggest especially to consider the paper elaborating an universal curve for apatite crystallinity (https://www.nature.com/articles/s41598-018-30642-z).

Thank you for your suggestion to reference the research of Gregorio dal Sasso, specifically dal Sasso et al. (2018). We are familiar with the work of dal Sasso and believe the universal curve of apatite crystallinity described in dal Sasso et al. (2018) is a significant recent advancement in the use of spectroscopy to evaluate the preservation of archaeological bone. We plan on using this methodology more extensively in our future work planned on the entire Tolbor-17 faunal assemblage, a comprehensive zooarchaeological study which is forthcoming.

I would also suggest to add (maybe in supplementary material) the section of Tolbon indicating the stratigraphic position of sampled bones. Please condor also that if bones come from different parts of the archaeological despots, they may have suffered different (I mean local) diagenetic process (e.g. related to water percolation or variations of pH values of the host archaeological layer); can this influence your results? the sole map of the study region is completely useless if you don't supply data on the stratigraphy of the site. Finally, it is not clear why you selected this specific archaeological sequence to carry out your experiments.

Thank you for your suggestion. As all burned faunal material sampled comes from a constrained area of one of the Tolbor-17 test pits and was considered to be firmly within the stratigraphic Unit 3, we do not believe this will be a complicating factor for our current study. We do plan on including more detailed sedimentological and geographic information in our more comprehensive zooarchaeological study of the full Tolbor-17 faunal assemblage, however, which will test our hypothesis of preservation bias highlighted by our conclusions further. We have clarified the stratigraphic position of the samples in our text to address this, however, and revised text can be found highlighted on page 13 of our manuscript with tracked changes.

We hope that we have also been able to clarify the inclusion of the Tolbor-17 samples in our study as initial investigations into the novel faunal preservation of Upper Paleolithic burned bone from the Tolbor valley, as well as representative samples of archaeological material sufficient to include as a reference sample to compare to our modern experimental samples. Text to acknowledge this in greater detail can be found highlighted on page 13 in our manuscript with tracked changes.

The description of methods is also very long and many informations are very basic. I suggest to shorten this part of the manuscript or move some parts to supplementary material.

We have taken your comment into serious consideration, and have tried to use more concise language to describe our methodologies. Ultimately much of the basic information have remained due to our interest in providing comprehensive detail to those reading our paper from diverse sub-disciplines who perhaps would like to recreate our study. 

Unfortunately, you do not present and discuss results of SEM investigation. Scanning microscope, I guess, may inform about the modification of apatite crystals of your samples; for instance changes in crystal shape or orientation, or can explain the evolution of diagenesis recording the process of recrystallization (evidence of dissolution and recrystallisation are very evident under the SEM). As you cited SEM in the Methods section, I think you should have interesting data to show. The only SEM images are in Fig. 7, but they are not discussed. SEM data would increase also the quality of discussion.

Thank you. For this study it was our original intention that the SE microscopy images be used for visualization purposes only, and have only selected five bones typical of their burning category for SE microscopy imaging (three from our modern experimental collection, and two from our archaeological assemblage). These bone samples were selected for imaging prior to being powdered for spectroscopic analyses, and it is no longer possible to image a more comprehensive sample of the other included burned modern and archaeological bone due to the powdering which took place. To address your suggestion, we have moved the detailed information regarding the SE microscopy instrument methodology to Supplemental Information and have clarified that the inclusion of SE images in this manuscript are for generalized visualization purposes only. In addition, greater detail of morphological observations are now included in the figure caption to address the processes which are present in our five images. We are excited for the suggestion to include a greater SE microscopy component in our future studies considering the thermal alteration of burnt bone, in which we will image a more thorough and comprehensive sample of material will be considered to conduct analyses and descriptions as you suggested.

---

## [Decision Letter · Decision Letter 1]

16 Mar 2021

PONE-D-20-19845R1

Characterization of structural changes in modern and archaeological burnt bone: implications for differential preservation bias

PLOS ONE

Dear Dr. Gallo,

Thank you for submitting your manuscript to PLOS ONE. After careful consideration, we feel that it has merit but does not fully meet PLOS ONE’s publication criteria as it currently stands. Therefore, we invite you to submit a revised version of the manuscript that addresses the points raised during the review process.

We look forward to receiving your revised manuscript.

Kind regards,

Justin W. Adams, Ph.D.

Academic Editor

PLOS ONE

Journal Requirements:

Additional Editor Comments (if provided):

On behalf of myself and the reviewers I would like to thank you for resubmitting your manuscript and addressing the minor and major points raised during the first peer review.

Having received a second round of reviews from the original reviewers, there are very few minor points which have been raised and should be easily addressed.

Based on my own independent evaluation, I agree with Reviewer 2 that the current Discussion could very easily be expanded. I appreciate and understand that there is the intent to provide a more expanded faunal analysis of the T17 sample in the future, nor does every individual publication have to include comprehensive coverage of all topics and content. However, given the general audience of PLOS One, it is equally critical to highlight to broader significance and importance of results. And as this presents novel data that has immediate relevance towards the archaeological sample under analysis, it would seem to be a significant missed opportunity to take these results on burnt bone and contextualise them in light of a larger consideration as to what significance this result will have to site-based interpretations. The introductory segment highlights (pages 7-8) the significance of the site for human biogeography, use of fire, the minimal archaeological/faunal record in the region, etc. Yet the Discussion (and separate Conclusions) fail to pick up on the significance of this locality and what the presented results can inform on in the planned future research. This lends the impression that the site itself, or the descriptions of results from the site, have no particular interpretative significance (and any site therefore could have been picked); and therefore is a missed opportunity to provide a reinforcement of the significance of the results for the T17 site in addition to the broader experimental outcomes. I don't view this as a onerous task but strongly recommend expanding the Discussion to revisit the site and how these results assist in framing understandings of the deposits (and their significance as alluded to in the first description of the materials).

Reviewers' comments:

Reviewer's Responses to Questions

**Comments to the Author**

1. If the authors have adequately addressed your comments raised in a previous round of review and you feel that this manuscript is now acceptable for publication, you may indicate that here to bypass the “Comments to the Author” section, enter your conflict of interest statement in the “Confidential to Editor” section, and submit your "Accept" recommendation.

Reviewer #1: All comments have been addressed

Reviewer #2: (No Response)

Reviewer #3: All comments have been addressed

2. Is the manuscript technically sound, and do the data support the conclusions?

Reviewer #1: Yes

Reviewer #2: Partly

Reviewer #3: Yes

3. Has the statistical analysis been performed appropriately and rigorously? 

Reviewer #1: N/A

Reviewer #2: N/A

Reviewer #3: Yes

4. Have the authors made all data underlying the findings in their manuscript fully available?

Reviewer #1: Yes

Reviewer #2: Yes

Reviewer #3: Yes

5. Is the manuscript presented in an intelligible fashion and written in standard English?

Reviewer #1: Yes

Reviewer #2: Yes

Reviewer #3: Yes

6. Review Comments to the Author

Reviewer #1: There are very minor changes suggested included in the pdf file attached. These refer to including a reference, be more precise in the sample size and delete a sentence from conclusions that is not really conclusion form the manuscript

Reviewer #2: Dear Authors,

I am quite surprised with the reviewed version of the manuscript. In my opinion, all the suggestions made by the reviewers were pertinent and most of them, apparently, easily realizable. But instead of reviewing and completing the manuscript with new information/discussion, it seems that the authors have choosen to rewrite and reorganize the text, it is better organized and better understood now, but they have not include new discussions or even explain the archaeological and stratigraphical context of archaeological remains analysed... or include sample´s photographs (as requested).

I feel that the analytical work and data is very OK, but the manuscript lacks a proper discussion. The nice analytical data support what already previous papers has published regarding the mineralogical, textural and compositional evolution of burned bone and it is a good starting point for a dscussion; but the discussion is just focused in comparing experimental results with results from archaeological data, it is OK, but which are the implications of having burnt bones in T17 ? Why it is important to have burned bones? How is archaeologically interpreted Unit 3?

In the same way, in the title "differential preservation bias" is mentioned, but it is not discussed in the text...which bias is detected in T17 collection? Why?

If the revisions of the reviewers are not tackled and the above mentioned comments corrected (they are easily realizable minor to moderate revisions) I think that this manuscript it is not suitable for PlosONE, it would be more appropiate for a more methodlogical/archaeometrical journal as e.g. Archaeometry or similar journals.

Reviewer #3: (No Response)

7. PLOS authors have the option to publish the peer review history of their article (what does this mean?). If published, this will include your full peer review and any attached files.

Reviewer #1: No

Reviewer #2: No

Reviewer #3: No

---

## [Author Response · Author response to Decision Letter 1]

13 Jun 2021

The following is copied from our Response to Reviewers document, uploaded as an attachment. 

Response to Reviewers:

Editorial comment: Based on my own independent evaluation, I agree with Reviewer 2 that the current Discussion could very easily be expanded... However, given the general audience of PLOS One, it is equally critical to highlight to broader significance and importance of results. 

Thank you for your constructive suggestions and encouragement to expand our Discussion. We agree that broadening our scope and contextualizing the inclusion of archaeological material results in a much stronger manuscript which speaks to a larger audience. 

And as this presents novel data that has immediate relevance towards the archaeological sample under analysis, it would seem to be a significant missed opportunity to take these results on burnt bone and contextualise them in light of a larger consideration as to what significance this result will have to site-based interpretations. 

Thank you. We have modified our Discussion portion of the manuscript to revisit the site of Tolbor-17 and expand into greater detail regarding the biogeography and relevance of studying fire in the Ikh-Tolborin-Gol, as well as highlighting the importance and relevance of the Tolbor-17 site to address larger questions of human technology.

Reviewer 1 There are very minor changes suggested included in the pdf file attached. These refer to including a reference, be more precise in the sample size and delete a sentence from conclusions that is not really conclusion form the manuscript.

 Thank you for your feedback and observations. The inclusion of van Hoesel et al. (2019) is an excellent contribution to our manuscript expanding on the experimental work regarding properties of combusted bone and a complement to the cited work of Reidsma et al. (2016). To address the faunal summary table, we have added number counts for the piece plotted material. We agree that the addition of this detail provides a more traditional representation of the zooarchaeology, and the quantification of the burnt bone fragments from screens will be a component of our future work on the Tolbor-17 fauna. Thank you, additionally, for your recommendation regarding the sentence addressing small bone fragments in our conclusion. We have adjusted our text to clarify the recovery of nearly all burnt bone fragments were from the screened material, and have moved the majority of that text to our Methodology section to avoid references in the Conclusion portion of our manuscript. 

Reviewer 2 …[The] discussion is just focused in comparing experimental results with results from archaeological data, it is OK, but which are the implications of having burnt bones in T17 ? Why it is important to have burned bones? How is archaeologically interpreted Unit 3? 

Thank you for your suggestion and useful perspective on expanding our discussion of the archaeology and the significance of fire presence at Tolbor-17. We have added a much larger section of our Discussion paragraphs to address this, including contextualizing Tolbor-17 within the Ikh-Tolberin-Gol, as well as the ultimate reasons we are interested in investigating Upper Paleolithic fire further. We have also detailed further the rare opportunity Tolbor-17 presents to study questions of this nature, as it is rare to have fire and faunal preservation in the valley.

 In the same way, in the title "differential preservation bias" is mentioned, but it is not discussed in the text...which bias is detected in T17 collection? Why?

Thank you. We have clarified our text to specify that the experimental work characterizing bioapatite growth and crystallinity (demonstrated in both modern and archaeological bone) highlights a vulnerability of charred bones that has yet to be described. Our conclusions therefore provide implications for biases in archaeological assemblages of burnt bone that may impact the visibility and study of fire presence and properties from a faunal perspective, and can be hypothesis-generating for future studies considering site specific burnt bone preservation.

---

## [Editor Report · Decision Letter 2]

28 Jun 2021

Characterization of structural changes in modern and archaeological burnt bone: implications for differential preservation bias

PONE-D-20-19845R2

Dear Dr. Gallo,

We’re pleased to inform you that your manuscript has been judged scientifically suitable for publication and will be formally accepted for publication once it meets all outstanding technical requirements.

Kind regards,

Justin W. Adams, Ph.D.

Academic Editor

PLOS ONE

Additional Editor Comments (optional):

Thank you for your careful consideration of the minor amendments that arose during your resubmission. I appreciate your attention to the comments from the reviewers and myself as part of that process. I'm happy to recommend acceptance of the submission.
---

## [Editor Report · Acceptance letter]

1 Jul 2021

PONE-D-20-19845R2 

Characterization of structural changes in modern and archaeological burnt bone: implications for differential preservation bias 

Dear Dr. Gallo:

I'm pleased to inform you that your manuscript has been deemed suitable for publication in PLOS ONE. Congratulations! Your manuscript is now with our production department. 

Kind regards, 

on behalf of

Dr. Justin W. Adams 

Academic Editor

PLOS ONE